# Activated Graphene Oxide-Calcium Alginate Beads for Adsorption of Methylene Blue and Pharmaceuticals

**DOI:** 10.3390/ma14216343

**Published:** 2021-10-23

**Authors:** Burcu Gunes, Yannick Jaquet, Laura Sánchez, Rebecca Pumarino, Declan McGlade, Brid Quilty, Anne Morrissey, Zahra Gholamvand, Kieran Nolan, Jenny Lawler

**Affiliations:** 1DCU Water Institute, School of Biotechnology, Dublin City University, Glasnevin, D09 NA55 Dublin, Ireland; burcu.gunes@dcu.ie (B.G.); declan.mcglade2@mail.dcu.ie (D.M.); brid.quilty@dcu.ie (B.Q.); z.gholamvand@gmail.com (Z.G.); 2Institut Technologie du vivant, University of Applied Sciences and Arts Western Switzerland, Rte de Moutier 14, 2800 Delémont, Switzerland; jaquet.yannick@gmail.com; 3Faculty of Biology, University of Oviedo Calle Catedrático Valentín Andrés Álvarez, 33006 Oviedo, Spain; lauusnchzbna@gmail.com (L.S.); rebecacgp@gmail.com (R.P.); 4DCU Water Institute, School of Mechanical and Manufacturing Engineering, Dublin City University, Glasnevin, D09 NA55 Dublin, Ireland; anne.morrissey@dcu.ie; 5DCU Water Institute, School of Chemical Sciences, Dublin City University, Glasnevin, D09 NA55 Dublin, Ireland; kieran.nolan@dcu.ie; 6Qatar Environment and Energy Research Institute (QEERI), Hamad Bin Khalifa University, Doha 34110, Qatar

**Keywords:** adsorption, graphene oxide, methylene blue, pharmaceuticals, kinetics, isotherms and thermodynamics

## Abstract

The remarkable adsorption capacity of graphene-derived materials has prompted their examination in composite materials suitable for deployment in treatment of contaminated waters. In this study, crosslinked calcium alginate–graphene oxide beads were prepared and activated by exposure to pH 4 by using 0.1M HCl. The activated beads were investigated as novel adsorbents for the removal of organic pollutants (methylene blue dye and the pharmaceuticals famotidine and diclofenac) with a range of physicochemical properties. The effects of initial pollutant concentration, temperature, pH, and adsorbent dose were investigated, and kinetic models were examined for fit to the data. The maximum adsorption capacities q_max_ obtained were 1334, 35.50 and 36.35 mg g^−1^ for the uptake of methylene blue, famotidine and diclofenac, respectively. The equilibrium adsorption had an alignment with Langmuir isotherms, while the kinetics were most accurately modelled using pseudo- first-order and second order models according to the regression analysis. Thermodynamic parameters such as ΔG°, ΔH° and ΔS° were calculated and the adsorption process was determined to be exothermic and spontaneous.

## 1. Introduction

Micropollutants such as pharmaceuticals, personal care products, surfactants and pesticides [1], as well as synthetic dyes [2], have been found virtually ubiquitously in environmental matrices over the past decade. One major source of organic micropollutants is the effluents from wastewater treatment plants (WWTPs), since most of these emerging contaminants are poorly biodegradable [3]. In addition, the hazards presented to human health and the ecosystem by thousands of trace contaminants in a “cocktail effect” are not yet well understood, although advances in effect-based biomonitoring aim to address this [4,5].

Methylene blue (MB) is a heterocyclic aromatic chemical dye used in textile, paper and cosmetic industries [6]. It is not highly toxic but has significant adverse impacts on aquatic ecosystems [2], retarding the photosynthetic activity of aquatic plants by affecting the light penetration, consuming dissolved oxygen or isolating metal ions, producing microtoxicity to organisms [7,8,9,10]. It can also be harmful to human health, causing heart rate increase, nausea and vomiting [11]. Methylene blue is widely used as an indicator pollutant to demonstrate the efficiency of novel adsorbent materials in the literature. Famotidine (FMTD) is a histamine H_2_^-^receptor antagonist used for treating gastroesophageal reflux disease and Zollinger–Ellison syndrome [12]. Famotidine has been shown to persist in WWTP effluents [13,14,15]. Diclofenac (DFC) is a non-steroidal anti-inflammatory drug prescribed to reduce inflammation, pain and dysmenorrhea; consumption is associated with serious dose-dependent gastrointestinal, renal and hepatic adverse effects, and increases vascular and coronary risks by about 33% [16,17]. Diclofenac is monitored in European surface waters under the watch list mechanism for the Water Framework Directive, and has been found almost ubiquitously in wastewater influent, wastewater effluent and surface waters [18]. Diclofenac exposure in trout has been shown to induce severe glomerulonephritis, resulting in kidney failure [19], and it has been implicated in the collapse of Asian vulture populations [20].

The removal of micropollutants and synthetic dyes using membrane-based technologies, ozonation, photolysis, photocatalysis [21,22,23], electrolysis, Fenton [24], photo-Fenton oxidation and electrochemical oxidation [25] has been extensively investigated in the literature. In particular, adsorption technology holds a lot of advantages, such as easy operation, fast decolorization and chemical oxygen demand removal efficiency; however, the main limitation of adsorption technology is the low and non-selective adsorption capacity of traditional adsorbents. The enhancement of adsorption capacity by increasing surface area and optimizing pore size has received much research attention [26,27,28,29,30,31].

Graphene-based materials have received increasing attention as potential candidates for composite preparation due to their high specific surface area and adsorption capacity. Graphene oxide (GO) is a two-dimensional complex of carbon atoms decorated with a multitude of oxygen-containing functional groups densely packed in a honeycomb framework [32]. GO has unique properties, such as a large theoretical surface area, high thermal and chemical stability, high conductivity and good mechanical flexibility [33], showing a great potential as an adsorbent for the removal of pharmaceuticals [34], heavy metals [35,36] or dyes [37]. However, using bare GO as an adsorbent agent causes the agglomeration of GO, which requires a complex high-speed centrifuge for GO separation from the solution [38]. Therefore, in this study, to stabilize the GO [39] and maximize the ease of recovery, GO was incorporated into an alginate matrix (an anionic polysaccharide used in paints, inks or pharmaceuticals). Alginate forms a hydrogel when mixed with divalent cations, such as Ca^2+^, giving good mechanical properties. Acid-activation of the beads provides an enhancement in the surface, area including micro- and mesopores [40,41]. In fact, the adsorption capacity of GO-montmorillonite/sodium alginate beads was recently investigated [42].

In this work, the acid-activated (0.1 M HCl pH 4) adsorption capacity of calcium alginate graphene oxide beads was evaluated as novel adsorbents for MB, FMTD and DFC removal. In addition, the influence of initial pollutant concentration, adsorbent dose, adsorption temperature and pH on adsorption capacity was investigated, along with an examination of the kinetic and thermodynamic modelling of the reactions.

## 2. Materials and Methods

### 2.1. Materials

Graphite flakes (GF) were purchased from Asbury Carbons. Diclofenac sodium (DFC, 99%), famotidine (FMTC), methylene blue (MB) and alginic acid sodium salt (Na-Alg) were purchased from Sigma Aldrich. The structure of analytes is given in Appendix B. Calcium chloride dihydrate, sodium hydroxide, potassium permanganate and absolute ethanol were purchased from Fischer Chemicals. Sulfuric acid (H_2_SO_4_, 95–98%) and hydrogen peroxide (H_2_O_2_, 30%) was purchased from Merck Millipore. Hydrochloric acid (37%) was provided by Acros Organics Dublin, Ireland.

### 2.2. Methods

#### Preparation of Graphene oxide (GO) Solution and Ca-Alg2/GO Beads

GO was prepared according to a modified version of Hummer’s method [43]. In this method, expanded graphite derived from graphite flasks is treated with H_2_SO_4_ to be mixed with H_2_O_2_ in order to produce GO particles. To establish the concentration, 1 g of GO suspension was placed in a dried, weighed beaker, dried overnight at 60 °C and weighed again, and the concentration was then adjusted to 1% GO in DI water on a dry mass basis. The details of the method and Ca-Alg2/GO bead preparation as well as dry and wet images of Ca-Alg and Ca-Alg/GO beads are given in Appendix C.

### 2.3. Acid Activation of the Beads Activation

Beads for activation were placed into 600 mL beakers of DI water adjusted to pH 4 using 0.1M HCl, which were agitated for 3 h. Afterwards, the beads were collected, rinsed three times with 300 mL of DI water and stored in a closed bottle at RT.

### 2.4. Characterization

The surface morphological structure of the beads was examined using scanning electron microscopy (SEM) analysis using a Hitachi 3400 SEM, following gold coating. Functional groups of the GO sheets as well as of the Ca-Alg2 and Ca-Alg2/GO beads were identified by Fourier transform infrared spectroscopy (Appendix D). In addition, the GO used for the beads’ preparation was characterized by X-ray diffraction and Raman spectroscopy (Appendix E).

### 2.5. Adsorption Measurements

All adsorption measurements were carried out in 250 mL flasks with 0.05 g of adsorbent (Ca-Alg2, Ca-Alg_2_/GO5, Ca-Alg2/GO10 or Ca-Alg2/GO20 dried beads) over 24 h on a shaker table operating at 125 rpm at room temperature (22 °C), unless otherwise specified. Equilibrium for all pollutants was established by 24 h. In total, 75 mL of pollutant solution at a concentration of 10 mg L^−^^1^ was added in each case, with a pH of 7 for MB and FMTD and a pH of 2 for DFC, unless otherwise specified. The pollutant concentration was determined using a UV-VIS spectrophotometer (Varian) at a wavelength of 660, 286 and 274 nm for MB, FMTD and DFC, respectively. Experiments were carried out in triplicate and the average values reported along with the error bars represent the standard deviation.

#### 2.5.1. Initial Pollutant Concentration

The initial pollutant concentrations tested were 10, 100, 500 and 1000 mgL^−1^ for MB, 10, 25, 100 and 250 mgL^−1^ for FMTD and 1, 5, 10 and 25 mgL^−1^ for DFC. The absorbed amount at equilibrium (q_eq_ (mg g^−1^)) was calculating using Equation (1):(1)$${\;q}_{\mathrm{eq}}=\frac{\left(C_{0}-C_{\mathrm{eq}}\right)\cdot V}{m_{\mathrm{ads}}}$$
where C_0_ (mg L^−1^) is the initial pollutant concentration, C_eq_ (mg L^−1^) the equilibrium pollutant concentration, V (L) the solution volume and m_ads_ (g) the adsorbent mass.

#### 2.5.2. Adsorbent Dose

The effect of the adsorbent dose was studied using 0.01, 0.025, 0.05 and 0.1 g of Ca-Alg2, Ca-Alg2/GO5, Ca-Alg2/GO10 or Ca-Alg2/GO20 dried beads.

#### 2.5.3. pH

The adsorption was performed at pH 7, 9, 10 and 11 for MB and FMTD whereas the adsorption for DFC was at pH 2, 3.5, 5 and 7.

#### 2.5.4. Temperature

The influence of the temperature was studied by performing the adsorption process at 4, 22 and 30 °C.

#### 2.5.5. Thermodynamics

The thermodynamic parameters of adsorption were determined at 4, 22 and 30 °C in order to evaluate the feasibility and the spontaneous nature of the adsorption. The adsorption distribution coefficient K_d_ is calculated using Equation (2):(2)$$K_{d}=\frac{C_{0}-C_{\mathrm{eq}}}{C_{\mathrm{eq}}}$$
where C_0_ (mg L^−^^1^) is the initial concentration of the solution and C_eq_ (mg L^−1^) the equilibrium concentration in solution. A plot of ln(K_d_) versus 1/T gives a straight line where the enthalpy change ΔH° (J·mol^−1^) and the entropy change ΔS° (J·K^−1^·mol^−1^) can be calculated using (Equation (3)):(3)$$\ln\left(K_{d}\right)=\frac{\Delta S{}^{\circ}}{R}-\frac{\Delta H{}^{\circ}}{\mathrm{RT}}$$
where R is the ideal gas constant (8.345 J·mol^−1^·K^−1^) and T (K) is the temperature of the solution during the adsorption process. The standard Gibbs free energy change can be obtained from Equation (4):(4)ΔG°=ΔH°− TΔS°

#### 2.5.6. Kinetics

Kinetic parameters were studied using 0.05 g of Ca-Alg2, Ca-Alg2/GO5, Ca-Alg2/GO10 or Ca-Alg2/GO20 dried beads. The three most common models were examined as to their fit to the experimental data [44]. The adsorbate capacity q_t_ (mg g^−1^) at time t was calculated using Equation (5):(5)$$q_{t}=\frac{\left(C_{0}-C_{t}\right)\cdot V}{m_{\mathrm{ads}}}$$
where C_0_ (mgL^−1^) is the initial concentration, C_t_ (mgL^−1^) the concentration at time t, V (L) the volume of pollutant solution and m_ads_ (g) the adsorbent mass.

The linearized integral form of the pseudo-first-order Lagergren equation is given by Equation (6):(6)$$\ln\left(q_{\mathrm{eq}}-{\;q}_{t}\right)=\ln\left(q_{\mathrm{eq}}\right)-k_{1}\cdot t$$
where k_1_ (min^−1^) is the Lagergren rate constant, q_eq_ (mg g^−1^) is the maximum adsorbed amount at equilibrium, and q_t_ (mg g^−1^) is the amount of adsorption at time t (min). The values of k_1_ and q_eq_ were determined from the intercept and the slope of a plot of ln(q_eq_ − q_t_) versus t.

The linearized integral form of the pseudo-second-order model is shown in Equation (7):(7)$$\frac{t}{q_{t}}=\frac{1}{k_{2}\cdot q_{\mathrm{eq}}^{2}}-\frac{1}{q_{\mathrm{eq}}}\cdot t$$
where k_2_ (g·mg^−1^·min^−1^) is the pseudo second-order rate constant of adsorption. The parameters k_2_ and q_eq_ were determined from the intercept and the slope of a plot of t/q_t_ versus t.

The intraparticle diffusion model is represented in Equation (8):(8)$$q_{t}={\;k}_{\mathrm{ip}}\cdot t^{1/2}+{\;C}_{\mathrm{ip}}$$
where k_ip_ (mg g^−1^·min^−0.5^) is an intraparticle diffusion rate constant and C_ip_ (mg g^−1^) is related to the thickness of the diffusion boundary layer. These parameters were obtained from a plot of q_t_ versus t^1/2^.

#### 2.5.7. Adsorption Isotherms

The Langmuir model and the Freundlich model were examined for their utility in describing the adsorption process. The Langmuir equation is detailed in Equation (9): [45]
(9)$$q_{\mathrm{eq}}={\;q}_{\max}\cdot \frac{C_{\mathrm{eq}}}{k_{L}+{\;C}_{\mathrm{eq}}}$$
where q_max_ (mg g^−1^) is the maximum adsorption capacity corresponding to complete monolayer coverage, C_eq_ (mgL^−1^) is the concentration at equilibrium in the solution and k_L_ (Lg^−1^) is a constant related to adsorption capacity and the energy of adsorption.

The Freundlich equation is an empirical model based on the adsorption on a heterogeneous surface [46], and is given in Equation (10):(10)$$q_{\mathrm{eq}}={\;k}_{F}\cdot C^{n}$$
where k_F_ (L·g^−1^) and n (-) are the Freundlich constants, indicating the adsorption capacity and the adsorption intensity, respectively. In order to determine the Langmuir and Freundlich constants, Excel Solver was used to fit the adsorption isotherm models with the experimental data. The sum of squared differences between experimental q_eq_ and calculated q_eq_ was minimized by changing the constants of the models with the solver to find the best non-linear regression.

### 2.6. Desorption Studies

After the concentration at equilibrium was determined, the beads were removed from the solution and were washed three times with DI water. The desorption of pollutants from the beads was examined using three different desorption systems, 0.1 M HCl, 1 M NaCl and ethanol 1% *v*/*v*. The desorption process was carried out in 250 mL conical flasks with 75 mL of desorption solution at RT. The conical flasks were agitated for 24 h at 125 rpm. Then, the final concentration in solution was determined using UV-VIS and the percentage desorption was calculated using Equation (11):(11)$$\mathrm{Desorption}\;=\frac{\left(q_{\mathrm{eq},a}-q_{\mathrm{eq},d}\right)}{q_{\mathrm{eq},a}}\cdot 100$$
where q_eq,d_ (mg g^−1^) is the adsorbed amount at equilibrium after 24 h of desorption, and q_eq,a_ (mg g^−1^) is the adsorbed amount at equilibrium after 24 h of adsorption.

## 3. Results and Discussion

### 3.1. Characterisation of Beads

SEM analysis was carried out in order to characterize the morphological structure of the beads, and the results are given Figure 1 and Figure 2 at 500× and 5000× magnification, respectively.

The SEM images show that increased graphene oxide concentration altered the morphological structure of the beads by providing increased porosity and roughness. Due to that increase, the beads had a greater surface available for interactions between adsorbate and adsorbent. The Ca-Alg2/GO20 was typically carbonaceous with similarities to the structure of activated carbon. The FTIR spectrum of GO sheets, Ca-Alg2 and Ca-Alg2/GO beads is given in S4. No significant difference was observed between Ca-Alg2 and Ca-Alg2/GO beads, as the functional groups of the alginate overlap with GO.

### 3.2. Effect of Operating Parameters on the Adsorption

#### 3.2.1. Contact Time

The effect of the contact time on q_t_ was examined by taking samples over 24 h. The average of the results obtained for the adsorption of MB, FMTD and DFC is given in Figure 3A–C, respectively.

The adsorption gradually increased with the contact time and slowed down progressively to reach an equilibrium after 24 h. An increased adsorbate capacity was achieved with an increased concentration of GO in the composites. Ca-Alg2/GO20 beads were the most efficient for the adsorption of each compound, as expected. For the adsorption of MB, Ca-Alg2/GO10 and Ca-Alg2/GO20 beads caused a significant improvement in the adsorbate capacity by increasing it from 6.91 ± 0.83 mg g^−1^ to 9.18 ± 0.08 and 10.63 ± 0.17 mg g^−1^, respectively, in comparison to Ca-Alg2 (control), with the *p* values of 0.034 and 0.023. For the adsorption of FMTD, regardless the level of GO, Ca-Alg2/GO composites showed a significant improvement (*p* < 0.05) in adsorbate capacity from 2.78 ± 0.34 mg g^−1^ to a maximum of 7.95 ± 0.54 mg g^−1^. On the other hand, for the adsorption of DFC, no significant enhancements were seen (high level of error bars), which is attributed to the molecular limitations of DFC.

#### 3.2.2. Pollutant Concentration

The effects of the different concentrations of methylene blue, famotidine and diclofenac on the adsorption density (q_eq_) and the percentage removal are shown in Figure 4, Figure 5 and Figure 6.

Higher adsorbed amounts of pollutant at equilibrium (q_eq_) were achieved with an increased GO level in the composites when the initial concentration of the pollutants (MB, FMTD and DFC) was higher. For the highest initial pollutant concentrations, which were 1000, 250 and 20 mgL^−1^ for MB, FMTD and DFC, respectively, all Ca-Alg2/GO beads (regardless of the GO level) showed a significant increase in q_eq_ value (*p* < 0.05) for DFC, whereas Ca-Alg2/GO10 and Ca-Alg2/GO20 beads caused a significant improvement in the q_eq_ value of MB and FMTD. The greatest enhancements were achieved with Ca-Alg2/GO20 beads, and were from 856.4 ± 16.8 to 1036.2 ± 30.7mg g^−1^ (*p*: 0.022) for MB, from 18.6 ± 1.9 to 32.2 ± 0.8 mg g^−1^ (*p*: 0.004) for FMTD and from 14.1 ± 0.329 to 20.4 ± 0.427 mg g^−1^ (*p*: 0.0001) for DFC at the highest pollutant concentrations.

Ca-Alg2/GO20 beads showed the highest adsorption percentage for each pollutant, as expected. The highest adsorption percentages were found to be 89.4 ± 0.25, 56.0 ± 1.7 and 80.9 ± 1.35 for MB, FMTD and DFC, respectively, which were significantly higher than the corresponding control.

#### 3.2.3. Adsorbent Dose

The effects of the adsorbent dose on adsorption density and adsorption percentage were observed by using four different masses of beads varying from 0.01 to 1.0. The adsorption densities (q_eq_) of four different types of beads and the percentages of adsorption of MB, FMTD and DFC on Ca-Alg2/GO20 beads are given in Figure 7, Figure 8 and Figure 9, respectively, as a function of the amount of adsorbent.

A decreased adsorption density has been observed with the increased adsorbent dose regardless of the type of beads and of pollutant. On the other hand, adsorption percentage increased significantly (*p* < 0.05) when using Ca-Alg2/GO20 beads from 52.9 ± 1.7 to 76 ± 0.4% for MB, from 12.2 ± 0.9 to 58.2 ± 0.5% for FMTD and from 39.7 ± 4.1 to 96.1 ± 1.7% for DFC, due to the increase in adsorbent dose from 0.1 to 1.0. On the contrary, the adsorption density decreased significantly (*p* < 0.05) from 38.9 ± 1.6 to 5.7 ± 0.1 mg g^−1^ for MB, from 9.2 ± 1.3 to 4.3 ± 0. mg g^−1^ for FMTD and from 29.5 ± 2.5 to 7.2 ± 0.1 mg g^−1^ for DFC due to the increase in adsorbent dose from 0.1 to 1.0. The reduction in the adsorption density was attributed to a lower quantity adsorbed per unit weight of the adsorbent, causing the presence of unsaturated adsorption sites [47] when the adsorbent dose was increased [48,49]. Ca-Alg2/GO20 beads were shown to exhibit significantly better adsorption than Ca-Alg2 beads.

#### 3.2.4. pH

The impact of the pH on adsorption was observed by using four pH values (7, 9, 10, and 11.5 for cationic molecules and 2, 3.5, 5 and 7 for anionic molecules). The averages of the results obtained for the adsorption of MB, FMTD and DFC are shown in the figures below (Figure 10A–C).

The adsorption percentage of MB at pH 7 was 62.9 ± 0.75%, which slightly increased up to 68.6 ± 0.7% at pH 10 and attained a maximum value (Figure 10A) when Ca-Alg2/GO20 beads were used. Similarly, the highest MB adsorptions were obtained at pH 10 with other beads. The adsorption percentage of FMTD showed a peak at pH 10 with for different types of beads, with the maximum of 49.2 ± 1.6% (Figure 10B) when Ca-Alg2/GO20 was used. Therefore, the adsorption of cationic molecules, MB and FMTD, increases with higher pH solutions until they reach a pH of 10, then it starts to decrease. This observation can be explained by looking at the pK_a_ values of the analytes and Ca-Alg2/GO20 beads. At pH 10, FMTD is in the neutral form, since its pK_a_ value is 6.98 [50], and it possess lower water solubility, thereby enhancing the adsorption process at this pH. MB still possesses a positive charge at pH 10; however, GO has an increased negative charge at pH 10 since the phenolic groups of GO now becoming ionized (GO pK_a_ = 4.3; 6.6; 9.8 all acid groups and 50% of GO phenolic groups will be ionized) [51], enhancing the charge attraction between MB and the adsorbent, which explains the larger adsorption capacity between Ca-Alg2/GO20 and Ca-Alg2 in Figure 10A.

On the other hand, for the anionic molecules, DFC, the highest adsorption percentage was observed as 96.1 ± 1.7% (Figure 10C) when Ca-Alg2/GO20 beads were used at pH 2. The adsorption percentage showed a sudden drop when the pH level increased to 3.5, and further increases in pH level had a negative impact on adsorption for four different bead types. The observed decrease in the adsorption of diclofenac at higher pH is a consequence of the non-ionized to ionized form of diclofenac (pK_a_ = 4.0), alginate (pK_a_ mannuronic = 3.38 and guluronic acid = 3.65 [52]) and GO. At pH 2, the ratio of non-ionized to ionized diclofenac is 100:1, that to alginate is 45:1 (pK_a_ 3.65), and that to GO is 200:1 (see Appendix F). These ratios change at pH 3.5 to 3.16:1, 1.4:1 and 6.3:1 for diclofenac, alginate and GO, respectively. Thus, diclofenac, alginate and GO gain negative charge, and as a consequence repulsion occurs. Furthermore, as diclofenac becomes negatively charged, its water solubility is significantly enhanced, consequently reducing adsorption [53,54]. The calculation of the pKa values of the pollutants and adsorbents is given in Appendix F.

#### 3.2.5. Temperature

Adsorption studies were performed across three temperatures ranging from 4 to 30 °C (4, 22 and 30 °C). The averages of the results obtained for the adsorption of MB, FMTD and DFC are shown in Figure 11A–C.

The coldest temperature (4 °C) showed the highest adsorption percentages of 65.3 ± 0.9% (Figure 11A), 53.3 ± 1.2% (Figure 11B) and 87.3 ± 3.4% (Figure 11C) for MB, FMTD and DFC, respectively when Ca-Alg2/GO20 beads were used. The adsorption of MB, FMTD and DFC decreased significantly due to stepwise increases in temperature to 22 and 30 °C, and it reached 61.1 ± 1.0, 38.6 ± 4.0 and 47.1 ± 2.0%, respectively, under the same conditions. This may be explained by an exothermic adsorption process [55].

### 3.3. Thermodynamics

Thermodynamic studies were conducted based on the feasibility and the spontaneous nature of adsorption [44]. The distribution coefficients for the adsorption K_d_, enthalpy change ΔH°, entropy change ΔS° and Gibbs free energy change ΔG° were calculated using Equations (2)–(4). The results are given in the Table 1, Table 2 and Table 3.

A significant decrease (*p* < 0.05) in the distribution coefficient (K_d_) was observed in all cases when the adsorption temperature was increased from 4 to 30 °C, indicating better adsorption at lower temperatures. For example, the K_d_ value for the adsorption of MB, FMTD and DFC on Ca-Alg2/GO20 beads decreased from 1.88 to 1.60, 1.14 to 0.63 and 6.88 to 0.93, respectively, as a result of increasing the temperature from 4 to 30 °C. Furthermore, negative enthalpy (ΔH°) and entropy (ΔS°) changes were seen in the adsorption of MB, FMTD and DFC on Ca-Alg2 and Ca-Alg2/GO beads (Table 1, Table 2 and Table 3). Negative enthalpy change indicates that the adsorption process is exothermic in nature, while negative entropy change suggests a reduction in randomness at the solid–solute interface during adsorption [56]. Moreover, the adsorption of MB and DFC on all beads was found to be spontaneous at 4 °C and 22 °C, respectively, due to the negative Gibbs free energy changes (ΔG°); however, the spontaneity decreased with the increasing temperature. Spontaneity was achieved when Ca-Alg2/GO20 beads were used as the adsorbent. The adsorption of FMDT on Ca-Alg2/GO20 was only found to be spontaneous at 4 °C. Several studies indicate that the absolute magnitude of the change in Gibbs free energy for physisorption is between −20 kJ·mol^−1^ and 0 kJ·mol^−1^, and chemisorption occurs between −80 kJ·mol^−1^ and −400 kJ·mol^−1^ [57,58]. Thus, the adsorption process observed seems to be physisorption.

### 3.4. Kinetics

Three models, pseudo-first-order Lagergren, the pseudo-second-order model and intraparticle diffusion, were fitted to the experimental data, and the models are given in Figure 12. All the kinetic parameters for the adsorption of MB, FMTD and DFC are given in Table 4, Table 5 and Table 6.

The adsorbed amounts of MB and FMTD predicted using the pseudo-first-order model are lower than the experimental data, and the values of R^2^ are better under the pseudo-second-order model. The experimental values for MB are 6.91 mg g^−1^ and 10.63 mg g^−1^ for Ca-Alg2 and Ca-Alg2/GO20, respectively, but the results for the pseudo-first-order model are 6.27 mg g^−1^ and 9.79 mg g^−1^ for Ca-Alg2 and Ca-Alg2/GO20, respectively; for the pseudo-second-order model, the results are 7.84 mg g^−1^ and 12.66mg g^−1^ higher than in the experimental data; however, the R^2^ values are 0.9907 instead of 0.9657 for the case of Ca-Alg2, and 0.9993 instead of 0.9991 for Ca-Alg2/GO20. For FMTD, the experimental results are 2.78 mg g^−1^ and 7.95 mg g^−1^, but the predictions of the pseudo-first-order model are 2.57 mg g^−1^ and 8.50 mg g^−1^ for Ca-Alg2 and Ca-Alg2/GO20, respectively; for the pseudo-second-order model, the predictions are 2.93 mg g^−1^ and 8.50 mg g^−1^; as we can see, the experimental data are higher than those from the pseudo-first-order model and lower than those from the pseudo-second-order model only for the Ca-Alg2 beads, while for the Ca-Alg2/GO20 beads the experimental results are lower than the predicted ones. However, the R^2^ results are better for the pseudo-second-order model. This indicates that the adsorption process does not fit the pseudo-first-order model, and shows applicability to the pseudo-second-order model for describing the adsorption of methylene blue and famotidine onto Ca-Alg2 and Ca-Alg2/GO beads [59].

The diffusion mechanism during the adsorption process was studied using the intraparticle diffusion model. The plot of q_t_ versus t^1/2^ shows a non-linear form, indicating that the adsorption process occurs in more than one step, as there are two distinct linear regions. The first straight region is attributed to macro-pore diffusion, and the second linear region to micro-pore diffusion. The first portion characterizes the instantaneous utilization of the adsorbing sites on the adsorbent surface. On the other hand, the second region is attributed to the slow diffusion of the methylene blue from the surface film into the micro-pores [60].

The predictions for the adsorbed amount of diclofenac obtained using the pseudo-first-order model fit the experimental data better than those obtained using the pseudo-second-order model. The experimental results are 3.78 mg g^−1^ and 5.81 mg g^−1^ for Ca-Alg2 and Ca-Alg2/GO20, respectively; for the pseudo-first-order model, the predictions are 3.70 mg g^−1^ and 4.64 mg g^−1^ for Ca-Alg2 and Ca-Alg2/GO20, respectively; for the pseudo-second-order model, the predictions are 4.26 mg g^−1^ and 6.33 mg g^−1^ for Ca-Alg2 and Ca-Alg2/GO20, respectively. The predictions are higher in every case except for the pseudo-first-order model and the Ca-Alg2/GO20 beads. Moreover, the values of the correlation coefficient R^2^ are higher for the pseudo-first-order model, meaning that the adsorption process of diclofenac into Ca-Alg2 and Ca-Alg2/GO beads can be described by the Lagergren model. The intraparticle diffusion model shows a straight line, indicating the adsorption process, because the intercept is close to 0 [61].

### 3.5. Adsorption Isotherms

The adsorption isotherms for methylene blue were built by testing nine different concentrations, namely, 1, 5, 10, 25, 50, 100, 250, 500 and 1000 mg·L^−1^. The adsorption was carried out over 24 h at 125 rpm, room temperature, and pH 7, with 0.05 g of beads. The isotherms obtained for each kind of bead are shown in Figure 13, and the data fit results are given in Table 7. The results show that an increase in GO concentration of the beads improves the adsorbed amount of dye at equilibrium. The Langmuir model fits the experimental data better than the Freundlich model, as indicated by the goodness-of-fit tests (Table 7).

The adsorption isotherms for FMTD were performed by using 1, 5, 10, 25, 50, 100 and 250 mgL^−1^ of solution, whereas for diclofenac, the concentrations were 1, 5, 10, 15 and 20 mgL^−1^. The adsorption process was carried out over 24 h at 125 rpm, room temperature, and pH 7, with 0.05 g of beads. The adsorption models of FMTD and DFC are given in Figure 14 and Figure 15, while the data fit results are given in Table 8 and Table 9, respectively.

The results for the adsorption isotherms of FMTD and DFC indicate similar behaviour to MB. Indeed, the Langmuir model better fits the experimental data between each compound and each different kind of bead. Table 7, Table 8 and Table 9 shows the Langmuir and Freundlich isotherm constants and correlation coefficients for MB, FMTD and DFC adsorption.

The values of R^2^ are higher with the Langmuir model, indicating that this model better fits the experimental data than the Freundlich isotherm for each kind of bead and each pharmaceutical. The Langmuir model presumes that the adsorption process occurs on a homogenous surface via monolayer adsorption.

The constants K_F_ and n indicate the adsorption capacity and the adsorption intensity, respectively. As indicated by the experimental data, the adsorption capacity K_F_ increases gradually with graphene oxide concentration, and the constant n is lower than 1, meaning the adsorption isotherm is favourable. The maximum adsorption capacities q_max_ obtained are 1334, 35, 50 and 36.35 mg g^−1^ for the uptake of methylene blue, famotidine and diclofenac, respectively. This means that Ca-Alg2/GO beads are an efficient adsorbent for the removal of these contaminants, particularly for methylene blue as, to our knowledge, this is the highest adsorption capacity for MB that has been reported in the literature.

### 3.6. Desorption

The desorption of each compound adsorbed onto Ca-Alg2 and Ca-Alg2/GO beads was studied using HCl/NaOH 0.1M, NaCl 1M and ethanol 1% v:v. The results of the percentage desorbed after 24 h are shown in Figure 16.

The desorption of MB from the beads was higher when using HCl 0.1 M than with NaCl or ethanol. The results show that 89 ± 9.9% and 44 ± 0.6% of MB are desorbed for Ca-Alg2 and Ca-Alg2/GO20 beads, respectively. Indeed, excesses of H^+^ protons seem to be able to force the cationic dye to be released by arising on the adsorption sites on the surfaces of the beads. It is more difficult to cause the release of MB from beads with graphene oxide due to the stronger affinity. NaCl 1 M also showed good results for the desorption of MB. However, the ionic strength of sodium chloride 1 M destabilizes the structure of calcium alginate beads, making them soft, fragile, and crumbly. As such, a high-concentration salt solution cannot be use as a desorbent due to the inability to reuse the beads. Ethanol solution showed low desorption of MB, as the main interactions between adsorbate and adsorbent are typically ionic bonds, and ethanol has a low ability to remove the dye from the beads with van der Waals’ forces.

On the other hand, the desorption of famotidine using HCl 0.1 M showed less satisfactory results. This might be due to the poor solubility in a low pH solution or to the presence of hydrogen bonds between the adsorbent and the adsorbate, making hydrochloric acid unable to release the drug into the solution. Hydrogen bonds could also explain why the ionic strength of NaCl is insufficient to remove the cationic pharmaceutical from the beads, in addition to destroying the stability of the beads. As for MB, ethanol has little effect on the van der Waals’ interaction for the removal of famotidine because of the stability of hydrogen bonds.

The results regarding the desorption of diclofenac from the beads showed that ethanol 1% v:v is able to release the pharmaceutical into solution. The percentages of desorption are high, around 70%, and the results present little difference between each of the kinds of beads. Diclofenac was desorbed from the beads with NaOH 0.1 M, mainly due to a change in the pH of the solution. As the results of the effect of pH showed, the adsorption of diclofenac was low in high pH solution. Similarly, to other compounds, NaCl is not an efficient desorbent, and it damages the stability of the beads by interacting with the structure of the polymer.

## 4. Conclusions

The SEM analysis showed that an increase in graphene oxide modified the morphological structure of the beads. Indeed, they become more porous and rougher with a higher surface available for the interactions between adsorbate and adsorbent. As expected, during the adsorption process, Ca-Alg2/GO20 beads present the best adsorption for each compound. The effect of initial concentration, adsorbent dose, pH and temperature all play an important role in the adsorption. The results show that a higher concentration of pharmaceuticals increases the force of the diffusion of the drug adsorbed by the beads. With a lower concentration of beads, the adsorbed amount at equilibrium q_eq_ increases because of a higher amount adsorbed per unit of weight of the adsorbent. On the other hand, the percentage of removal decreases due to the fewer adsorption sites available. The pH can modify the structure of the beads along with the pharmaceuticals, causing a change in the interactions between adsorbate and adsorbent. The adsorption process is better at low temperature than high temperature, meaning the adsorption mechanism is exothermic (as confirmed by H°), and thermodynamic studies show that the physisorption is spontaneous. The pseudo-second-order model is the best fit for the experimental data concerning the kinetics of adsorption for methylene blue and famotidine; however, the Lagergren pseudo-first order model is better suited to diclofenac. Furthermore, each compound follows the Langmuir model for isotherm adsorption, with a maximum adsorption capacity of 1334, 35.50 and 36.35 mg·g^−1^ for methylene blue, famotidine and diclofenac, respectively. It should be noted that the adsorption capacity of MB onto calcium alginate graphene oxide beads, particularly Ca-Alg2/GO20 composites, was found to be superior in comparison to other adsorbents, ranging from commercial activated carbon adsorption capacity (980.3 mg·g^−1^) to bioadsorbents adsorption capacity, such as modified biomass of baker’s yeast (869.6 mg·g^−1^), and many other natural adsorbents, such as biomass and coal-derived activate carbon, as reviewed in detail by [62]. In addition, GO and alginate incorporation outperformed a graphene-derived nanocomposite (Fe_3_O_4_^-^ graphene at mesoporous SiO_2_), wherein the adsorption capacity of this graphene-derived nanocomposite was reported to be 178.49 mg·g^−1^ in terms of MB removal [32]. Furthermore, the adsorption capacity of MB was recently published as 150.66 mg·g^−1^ when GO was incorporated with sodium alginate to produce aero gel beads. Therefore, it can be concluded that graphene oxide calcium alginate composite is a superabsorbent useful for MB removal from water, in addition to being superior to the previously researched compounds [42,63]. Furthermore, by treating the Alg2/GO20 composites that had come in contact with the absorbates MB or diclofenac with 0.1 M HCl and ethanol 1% v:v, the absorbates could be efficiently removed/desorbed, and the Alg2/GO20 composite beads were regenerated without damage to bead integrity. Further investigations need to be performed for famotidine, as little desorption was observed with the desorption candidates examined.

These beads appear to be an efficient adsorbent for dyes and pharmaceuticals, particularly for methylene blue. This novel technology could be applied as a polishing step in water treatment in order to reduce the concentration of these micropollutants, as well as of the synthetic dyes that negatively impact the environment, human health and aquatic life. Methylene blue is widely used a model pollutant in adsorption studies. It is interesting to note that while the performance of the beads assessed in this study is excellent for MB, it is less than ideal for FMTD and DFC. This calls into question the validity of the common approach of using a single-component pollutant for novel adsorbent testing. However, the treatment of an MB, FMTD and DFC mixture via Alg2/GO beads might have an influence on the adsorption/desorption capacities. Therefore, conducting an experimental study targeting the removal of mixed pollutants via Alg2/GO beads is suggested as a future direction.

## Figures and Tables

**Figure 1 materials-14-06343-f001:**
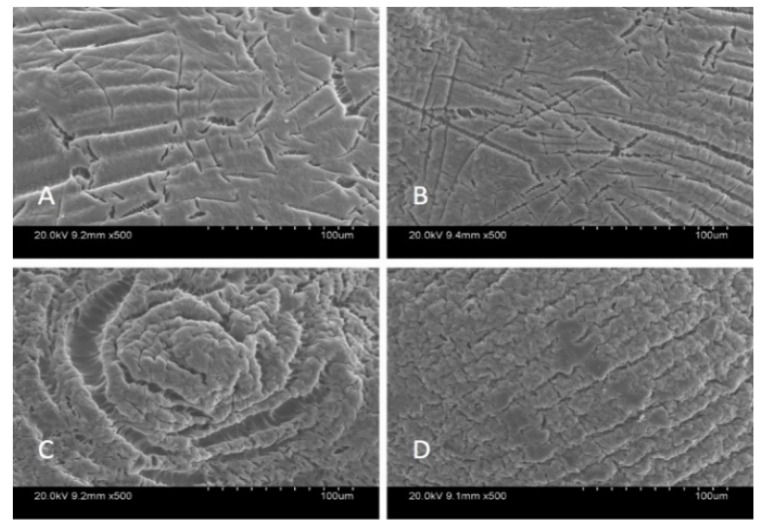
SEM images of beads at 500× magnification: Ca-Alg2 (**A**), Ca-Alg2/GO5 (**B**), Ca-Alg2/GO10 (**C**), Ca-Alg2/GO20 (**D**).

**Figure 2 materials-14-06343-f002:**
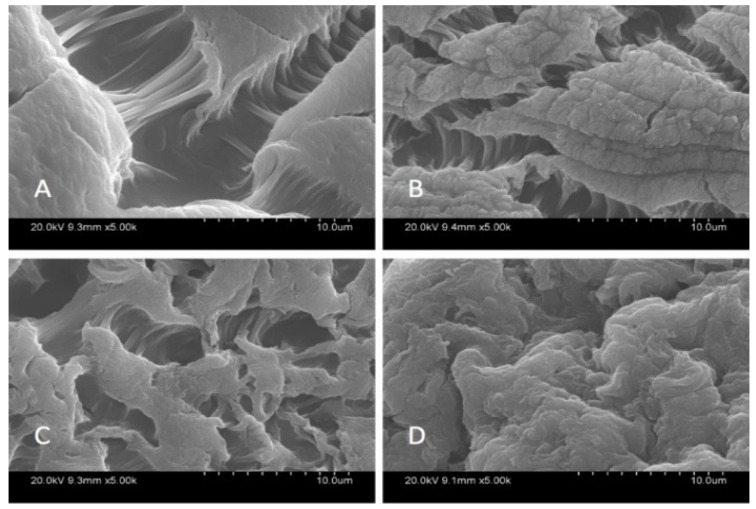
SEM images of beads at 5000×; Ca-Alg2 (**A**), Ca-Alg2/GO5 (**B**), Ca-Alg2/GO10 (**C**), Ca-Alg2/GO20 (**D**).

**Figure 3 materials-14-06343-f003:**
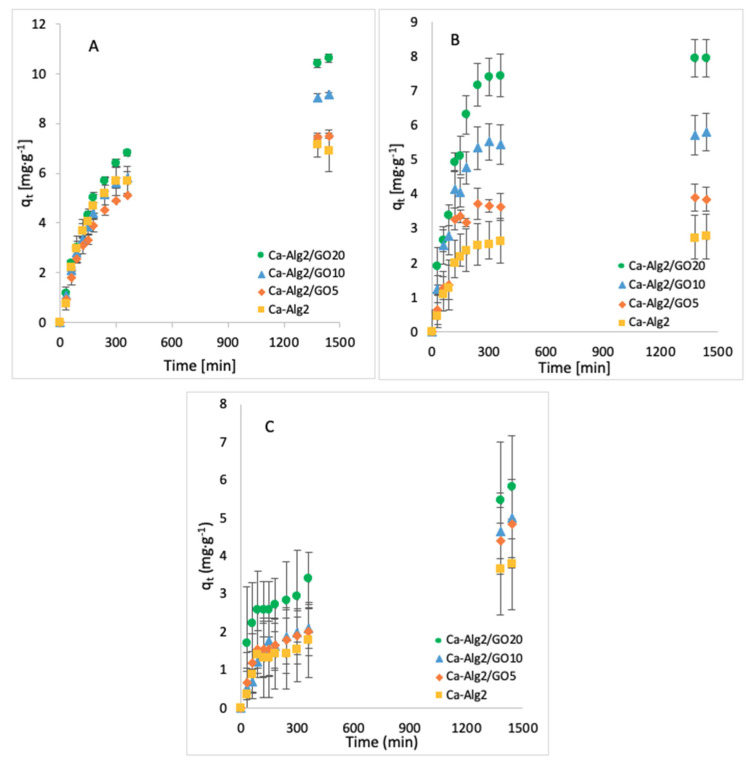
Effect of the contact time on q_eq_ of MB (**A**), FMTD (**B**) and DFC (**C**).

**Figure 4 materials-14-06343-f004:**
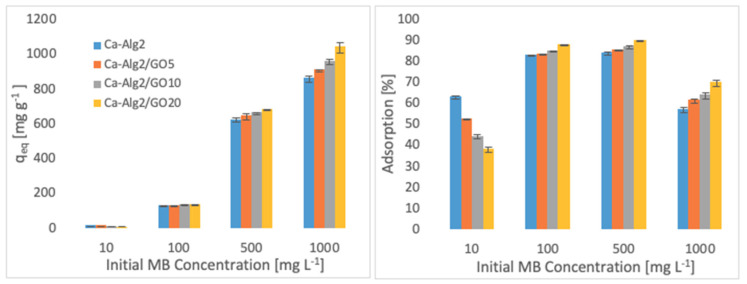
Effect of the initial MB concentration on q_eq_ and adsorption percentage.

**Figure 5 materials-14-06343-f005:**
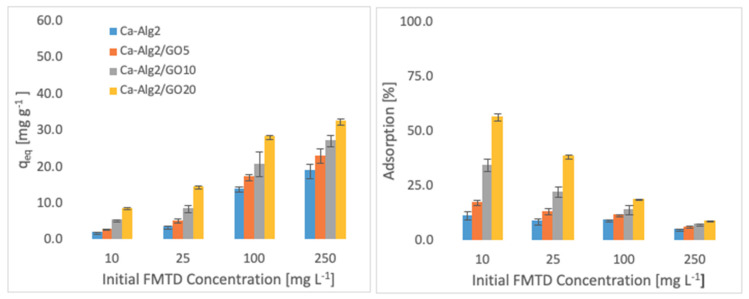
Effect of the initial concentration of FMTD on q_eq_ and adsorption percentage.

**Figure 6 materials-14-06343-f006:**
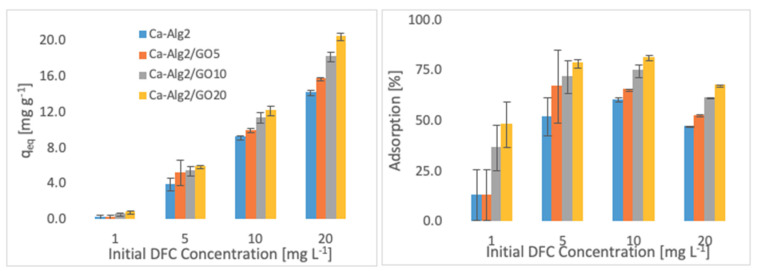
Effect of the initial concentration of DFC on q_eq_ and adsorption percentage.

**Figure 7 materials-14-06343-f007:**
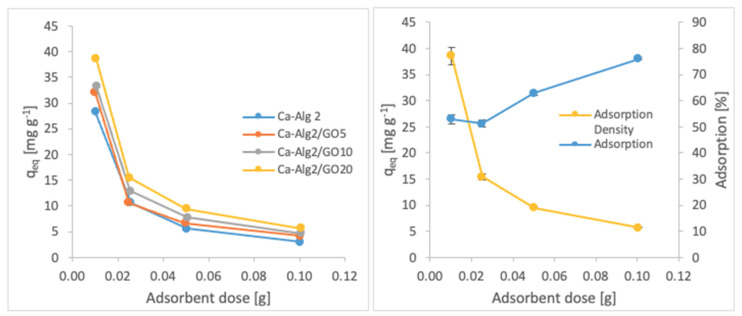
Effect of the adsorbent dose on qeq MB.

**Figure 8 materials-14-06343-f008:**
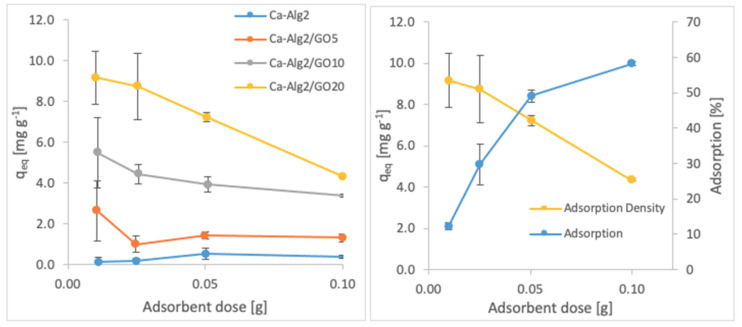
Effect of the adsorbent dose on qeq FMTD.

**Figure 9 materials-14-06343-f009:**
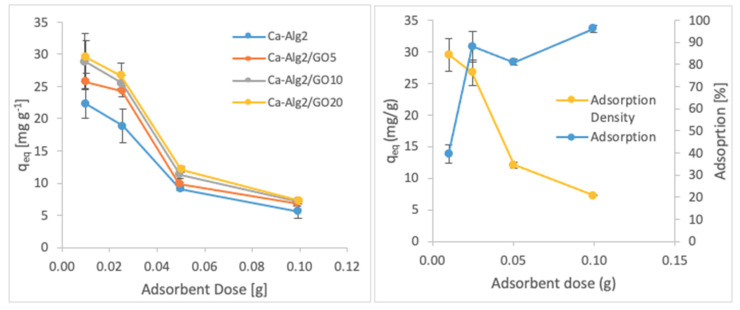
Effect of the adsorbent dose on qeq DFC.

**Figure 10 materials-14-06343-f010:**
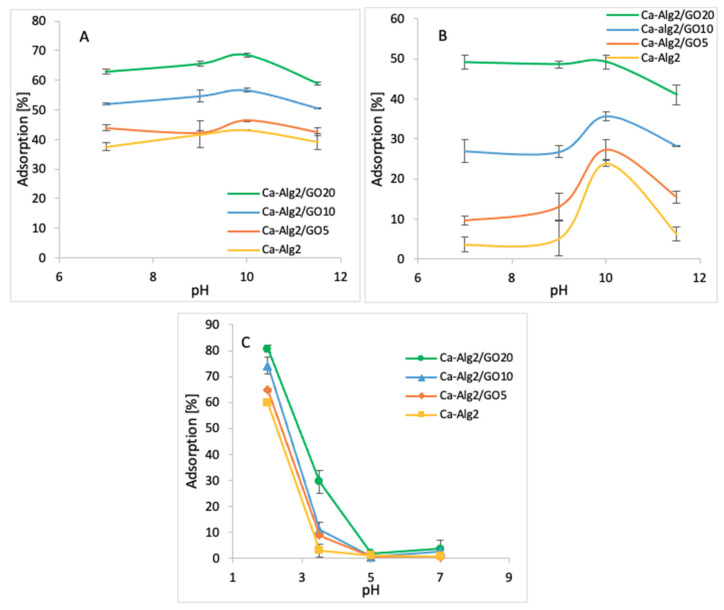
Effect of the pH on q_eq_ MB (**A**), FMTD (**B**), DFC (**C**).

**Figure 11 materials-14-06343-f011:**
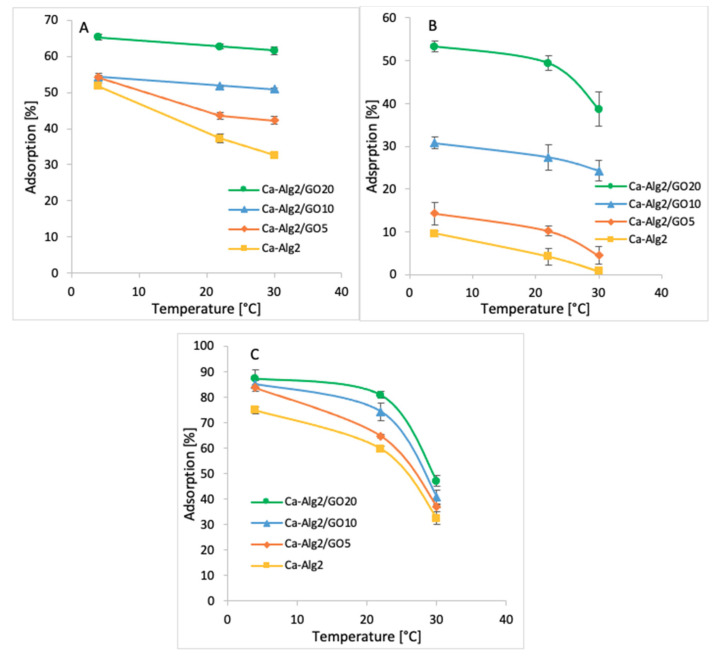
Effect of the temperature on q_eq_ of MB (**A**), FMTD (**B**), DFC (**C**).

**Figure 12 materials-14-06343-f012:**
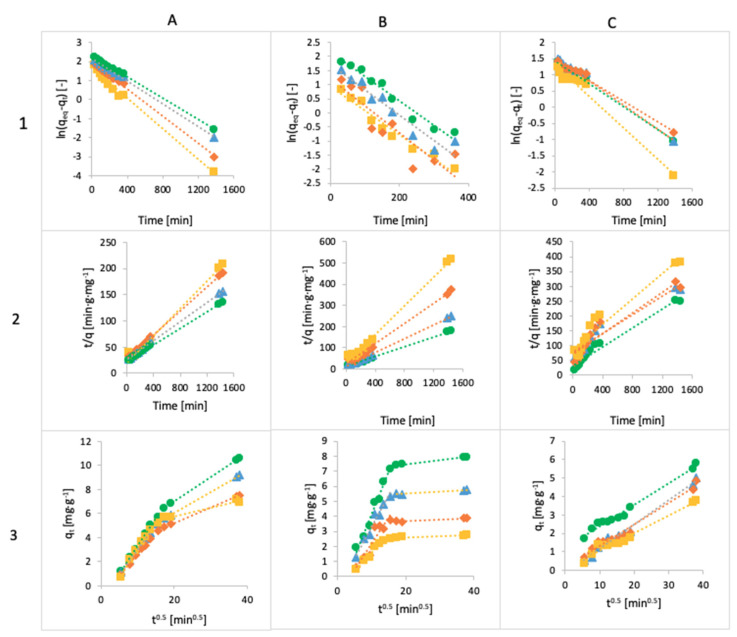
The linearized integral form of the pseudo-first-order Lagergren equation of methylene blue (**A1**), famotidine (**B1**) and diclofenac (**C1**), linearized integral form of the pseudo-second-order model of methylene blue (**A2**), famotidine (**B2**) and diclofenac (**C2**) and intraparticle diffusion model of methylene blue (**A3**), famotidine (**B3**) and diclofenac (**C3**) using Ca-Alg2 (◼), Ca-Alg2/GO5 (♦), Ca-Alg2/GO10 (▲) and Ca-Alg2/GO20 (●) beads.

**Figure 13 materials-14-06343-f013:**
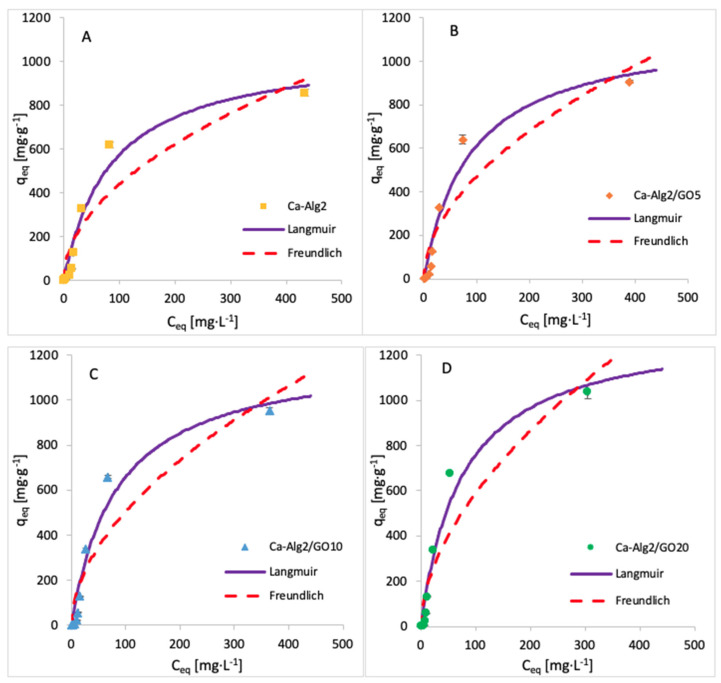
Adsorption isotherms for MB using (**A**) Ca-Alg2, (**B**) Ca-Alg2/GO5, (**C**) Ca-Alg2/GO10 and (**D**) Ca-Alg2/GO20 dried beads.

**Figure 14 materials-14-06343-f014:**
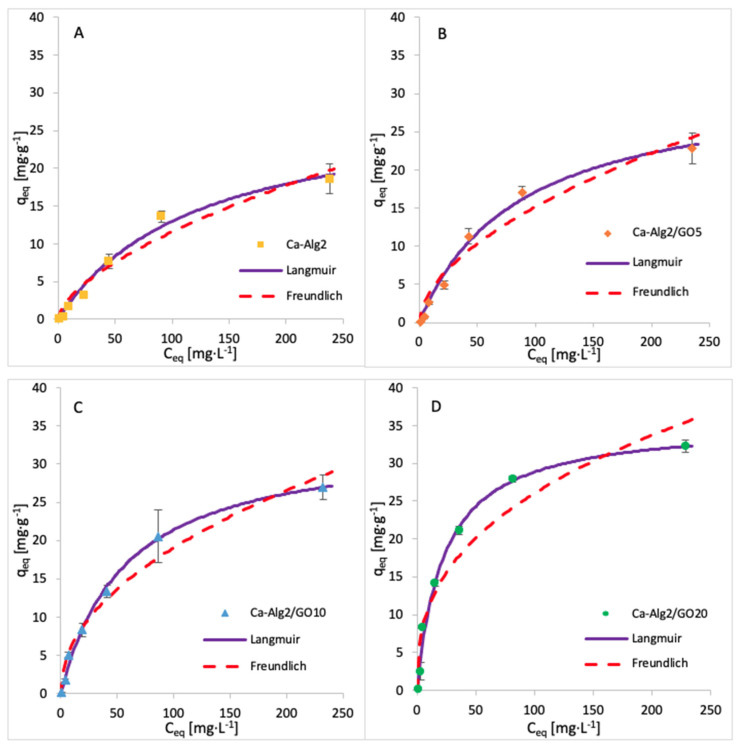
Adsorption isotherms of FMTD using (**A**) Ca-Alg2, (**B**) Ca-Alg2/GO5, (**C**) Ca-Alg2/GO10 and (**D**) Ca-Alg2/GO20 dried beads.

**Figure 15 materials-14-06343-f015:**
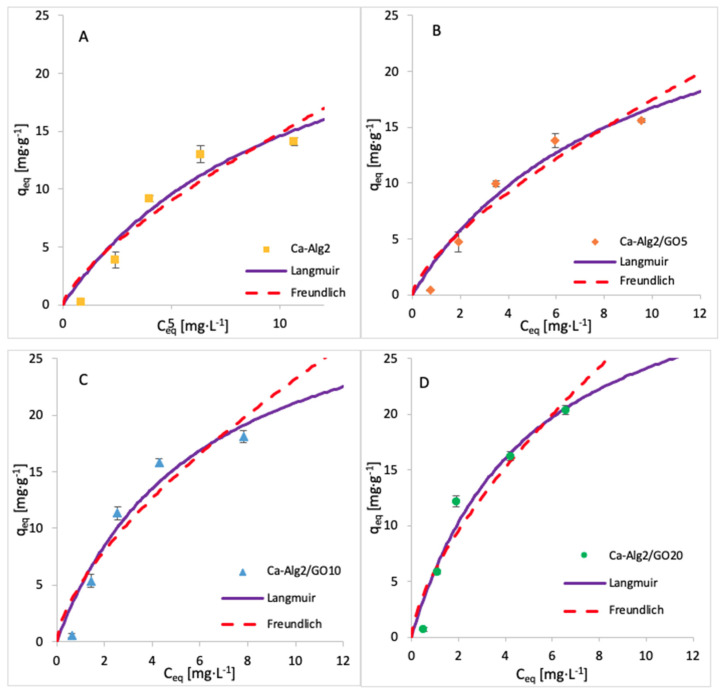
Adsorption isotherms of DFC using (**A**) Ca-Alg2, (**B)** Ca-Alg2/GO5, (**C**) Ca-Alg2/GO10 and (**D**) Ca-Alg2/GO20 dried beads.

**Figure 16 materials-14-06343-f016:**
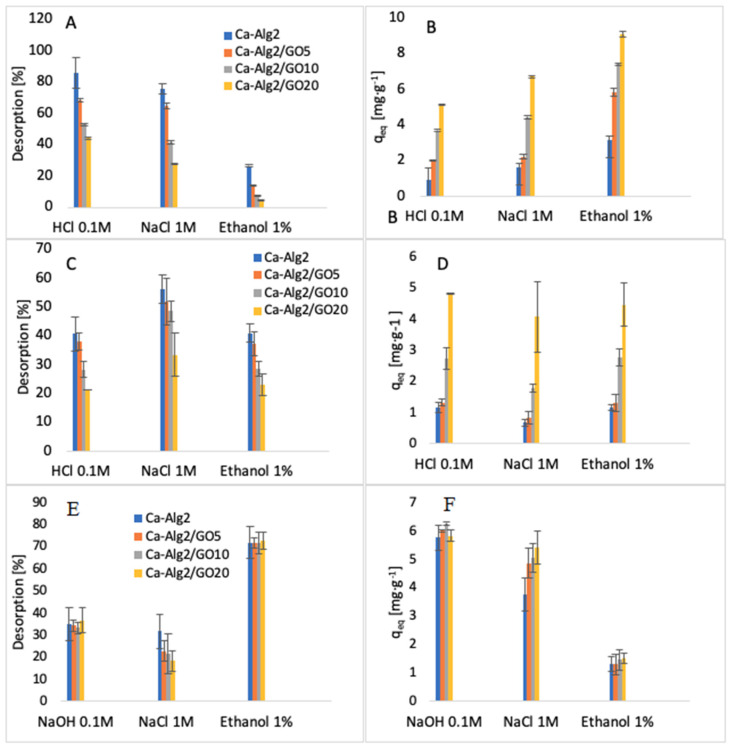
Desorption (%) of MB (**A**), desorption q_eq_ of MB (**B**), desorption (%) of FMTD (**C**), desorption q_eq_ of FMTD (**D**), desorption (%) of DFC (**E**) and desorption q_eq_ of DFC (**F**).

**Table 1 materials-14-06343-t001:** Thermodynamic parameters for MB adsorption on Ca-Alg2, Ca-Alg2/GO5, Ca-Alg2/GO10 and Ca-Alg2/GO20 beads.

Adsorbent	K_d_			ΔG° (kJ·mol^−1^)			ΔH° (kJ·mol^−1^)	ΔS° (J·K^−1^·mol^−1^)
	4 °C	22 °C	30°C	4 °C	22 °C	30 °C		
Ca-Alg2	1.07	0.59	0.48	−0.144	1.240	1.855	−21.44	−76.9
Ca-Alg2/GO5	1.18	0.77	0.73	−0.341	0.504	0.879	−13.34	−46.9
Ca-Alg2/GO10	1.19	1.08	1.04	−0.401	−0.186	0.091	−3.712	−12.0
Ca-Alg2/GO20	1.88	1.68	1.60	−1.459	−1.272	−1.189	−4.338	−10.4

**Table 2 materials-14-06343-t002:** Thermodynamic parameters for FMTD adsorption on Ca-Alg2, Ca-Alg2/GO5, Ca-Alg2/GO10 and Ca-Alg2/GO20 beads.

Adsorbent	K_d_			ΔG° (kJ·mol^−1^)			ΔH° (kJ·mol^−1^)	ΔS°(J·K^−1^·mol^−1^)
	4 °C	22 °C	30 °C	4 °C	22 °C	30 °C		
Ca-Alg2	0.11	0.04	0.01	4.747	9.465	11.56	−67.85	−262
Ca-Alg2/GO5	0.17	0.11	0.05	3.926	6.226	7.248	−31.46	−128
Ca-Alg2/GO10	0.44	0.38	0.32	1.847	2.505	2.797	−8.277	−36.6
Ca-Alg2/GO20	1.14	0.98	0.63	−0.418	0.473	0.868	−14.13	−49.5

**Table 3 materials-14-06343-t003:** Thermodynamic parameters for DFC adsorption on Ca-Alg2, Ca-Alg2/GO5, Ca-Alg2/GO10 and Ca-Alg2/GO20 beads.

Adsorbent	K_d_			ΔG° (kJ·mol^−1^)			ΔH° (kJ·mol^−1^)	ΔS° (J·K^−1^·mol^−1^)
	4 °C	22 °C	30 °C	4 °C	22 °C	30 °C		
Ca-Alg2	2.90	1.55	0.55	−2.758	−0.042	1.165	−44.56	−151
Ca-Alg2/GO5	4.96	1.80	0.58	−3.982	−0.704	0.753	−54.43	−182
Ca-Alg2/GO10	5.48	2.61	0.74	−4.352	−1.342	−0.004	−50.67	−167
Ca-Alg2/GO20	6.88	3.81	0.93	−4.896	−2.047	−0.781	−48.74	−158

**Table 4 materials-14-06343-t004:** Kinetic parameters for MB adsorption onto Ca-Alg2, Ca-Alg2/GO5, Ca-Alg2/GO10 and Ca-Alg2/GO20 dried beads.

Adsorbent	Pseudo-First-Order			Pseudo-Second-Order			Intraparticle Diffusion		
	q_eq_ (mg g^−1^)	k_1_ (min^−1^)	R^2^ (-)	q_eq_ (mg g^−1^)	k_2_ (g·mg^−1^·min^−1^)	R^2^ (-)	k_id_ (mg g^−1^·min^−0.5^)	C (mg g^−1^)	R^2^ (-)
Ca-Alg2	6.27	5.1 × 10^−3^	0.9657	7.84	8.3 × 10^−4^	0.9907	0.479	−1.68	0.9580
Ca-Alg2/GO5	7.22	3.6 × 10^−3^	0.9974	8.70	5.0 × 10^−4^	0.9995	0.358	−0.957	0.9997
Ca-Alg2/GO10	8.55	3.0 × 10^−3^	0.9979	10.81	3.5 × 10^−4^	0.9991	0.409	−1.10	0.9996
Ca-Alg2/GO20	9.79	2.8 × 10^−3^	0.9991	12.66	2.7 × 10^−4^	0.9993	0.457	−1.28	0.9998

**Table 5 materials-14-06343-t005:** Kinetic parameters for FMTD adsorption onto Ca-Alg2, Ca-Alg2/GO5, Ca-Alg2/GO10 and Ca-Alg2/GO20 dried beads.

Adsorbent	Pseudo-First-Order			Pseudo-Second-Order			Intraparticle Diffusion		
	q_eq_ (mg g^−1^)	k_1_ (min^−1^)	R^2^ (-)	q_eq_ (mg g^−1^)	k_2_ (g·mg^−1^·min^−1^)	R^2^ (-)	k_id_ (mg g^−1^·min^−0.5^)	C (mg g^−1^)	R^2^ (-)
Ca-Alg2	3.70	2.4 × 10^−3^	0.9756	4.64	5.5 × 10^−4^	0.9405	0.248	−0.882	0.8809
Ca-Alg2/GO5	4.28	1.6 × 10^−3^	0.9820	5.72	4.4 × 10^−4^	0.9147	0.340	−1.23	0.9355
Ca-Alg2/GO10	4.74	1.9 × 10^−3^	0.9865	6.31	3.3 × 10^−4^	0.9422	0.409	−0.821	0.8817
Ca-Alg2/GO20	4.26	1.8 × 10^−3^	0.9879	6.33	7.7 × 10^−4^	0.9712	0.556	−1.44	0.9992

**Table 6 materials-14-06343-t006:** Kinetic parameters for DFC adsorption onto Ca-Alg2, Ca-Alg2/GO5, Ca-Alg2/GO10 and Ca-Alg2/GO20 dried beads.

Adsorbent	Pseudo-First-Order			Pseudo-Second-Order			Intraparticle Diffusion		
	q_eq_ (mg g^−1^)	k_1_ (min^−1^)	R^2^ (-)	q_eq_ (mg g^−1^)	k_2_ (g·mg^−1^·min^−1^)	R^2^ (-)	k_id_ (mg g^−1^·min^−0.5^)	C (mg g^−1^)	R^2^ (-)
Ca-Alg2	3.70	2.4 × 10^−3^	0.9756	4.64	5.5 × 10^−4^	0.9405	0.095	0.107	0.9684
Ca-Alg2/GO5	4.28	1.6 × 10^−3^	0.9820	5.72	4.4 × 10^−4^	0.9147	0.117	0.110	0.9740
Ca-Alg2/GO10	4.74	1.9 × 10^−3^	0.9865	6.31	3.3 × 10^−4^	0.9422	0.131	−0.130	0.9858
Ca-Alg2/GO20	4.26	1.8 × 10^−3^	0.9879	6.33	7.7 × 10^−4^	0.9712	0.117	1.190	0.9835

**Table 7 materials-14-06343-t007:** Langmuir and Freundlich isotherm constants for MB adsorption onto Ca-Alg2 and Ca-Alg2/GO beads.

Adsorbent	Langmuir			Freundlich		
	q_max_ (mg·g^−1^)	K_L_ (L·g^−1^)	R^2^ (-)	K_F_ (L·g^−1^)	n (-)	R^2^ (-)
Ca-Alg2	1064	85.56	0.9778	43.26	0.503	0.9270
Ca-Alg2/GO5	1153	88.93	0.9716	41.02	0.530	0.9109
Ca-Alg2/GO10	1212	84.21	0.9782	42.64	0.537	0.8941
Ca-Alg2/GO20	1334	76.21	0.9894	45.10	0.558	0.8541

**Table 8 materials-14-06343-t008:** Langmuir and Freundlich isotherm constants for FMTD adsorption onto Ca-Alg2 and Ca-Alg2/GO beads.

Adsorbent	Langmuir			Freundlich		
	q_max_ (mg·g^−1^)	K_L_ (L·g^−1^)	R^2^ (-)	K_F_ (L·g^−1^)	n (-)	R^2^ (-)
Ca-Alg2	28.96	123.2	0.9809	0.680	0.615	0.9351
Ca-Alg2/GO5	31.69	85.39	0.9733	1.190	0.552	0.9173
Ca-Alg2/GO10	33.57	57.02	0.9611	2.099	0.479	0.8593
Ca-Alg2/GO20	35.50	23.10	0.9214	4.647	0.374	0.7491

**Table 9 materials-14-06343-t009:** Langmuir and Freundlich isotherm constants for DFC adsorption onto Ca-Alg2 and Ca-Alg2/GO beads.

Adsorbent	Langmuir			Freundlich		
	q_max_ (mg·g^−1^)	K_L_ (L·g^−1^)	R^2^ (-)	K_F_ (L·g^−1^)	n (-)	R^2^ (-)
Ca-Alg2	30.74	11.10	0.9457	2.795	0.725	0.8937
Ca-Alg2/GO5	31.81	9.020	0.9175	3.441	0.705	0.8707
Ca-Alg2/GO10	33.72	5.988	0.8886	5.055	0.662	0.8401
Ca-Alg2/GO20	36.35	5.066	0.8872	5.992	0.672	0.8322

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
