# Peer review of "Activated Graphene Oxide-Calcium Alginate Beads for Adsorption of Methylene Blue and Pharmaceuticals"

_materials, 2021, doi:10.3390/ma14216343_

Round 1
Reviewer 1 Report
- The paper is an adsorption study of a composite material based on calcium alginate for water treatment applications. The alginate was mixed with increasing content of graphene oxide (GO) and the effect of the GO content and several operating parameters on the adsorption where studied.
- The article is extensive, reads well, and is well structured. As a strength, the paper has real statistical study and base their conclusion on thermodynamic data.
- I found a few typos: qeq that is missing the subscript in several figure legends. “chance” instead of “change” in line 389.
- To reinforce the paper: The author state in L64 “however the main limitation of adsorption technology is the low and non-selective adsorption capacity of traditional adsorbents.” The authors do not talk about selectivity and although they mention their beads have the highest adsorption capacity for MB they do not compare with literature (an additional graph of table would be of interest here).
- Figure 12 should be improved. The reader can not appreciate the fit. There is too many points grouped together.
- Figure 16 should be improved to be more similar to Figure 6.
Author Response
Authors sincerely thank to the reviewer for his/her valuable comments to enhance the quality of our manuscript.

Reviewer 2 Report
In this work, the authors performed a detailed study to evaluate the absorption performance of pollutants by using calcium alginate graphene oxide composites and further investigated various influential factors with the examination of kinetic and thermodynamic modeling of the reactions. The prepared beads appeared to be an efficient adsorbent for dyes and pharmaceuticals. The results were presented clearly and analyzed comprehensively. However, there are a few issues that should be addressed before publishment.
- GO has been widely applied in pollutant absorption. The authors presented the surpassed absorption capacity of Ca-Alg2/GO beads compared with Ca-Alg2. But the absorption performance is suggested to be further compared with bare GO to illustrate the necessity of adding Ca-Alg2.
- It’s hard to observe GO in the SEM images. It is suggested that taking SEM, TEM, or AFM images of bare GO without Ca-Alg2 to characterize the morphologies of GO (size, thickness). The results can be attached in the Appendix.
- The authors have studied the absorption of MB, FMTD, and DFC, respectively. But the absorbent will be applied in mixed systems, practically. Would there be any interactions among different pollutants and the absorbent? What will be the absorption performance of Ca-Alg2/GO beads dealing with mixed pollutants?
Author Response

(The authors gave the same response as above.)
